

# The influence of physical activity levels on lactate production during squat training using a functional electromechanical dynamometer

Indya del-Cuerpo[1,2], Daniel Jerez-Mayorga[1,2,3], Luis Javier Chirosa-Ríos[1,2], Felipe Caamaño-Navarrete[4] and Pedro Delgado-Floody[5]

[1] Department of Physical Education and Sport, Faculty of Sports Sciences, University of Granada, Granada, Granada, Spain
[2] Strength & Conditioning Laboratory, CTS-642 Research Group, Department Physical Education and Sports, Faculty of Sport Sciences, University of Granada, Granada, Granada, Spain
[3] Exercise and Rehabilitation Sciences Institute, School of Physical Therapy, Faculty of Rehabilitation Sciences, Universidad Andrés Bello, Santiago, Santiago, Chile
[4] Physical Education Career, Universidad Autónoma de Chile, Temuco, Temuco, Chile
[5] Department of Physical Education, Sport and Recreation, Universidad de La Frontera, Temuco, Temuco, Chile

Corresponding author
Pedro Delgado-Floody,
pedro.delgado@ufrontera.cl

## ABSTRACT

**Objective**. This study aimed to determine the association between changes in lactate production and levels of physical activity in a group of healthy young adults in response to two squat training protocols.

**Material and methods**. Twenty-nine students majoring in Sports Science willingly participated in this study. Participants visited the lab four times within a two-week period, ensuring at least 48 h between visits. In each session, they completed three sets of 12 repetitions at 75% 1RM and three sets of 30 repetitions at 50% of maximum strength, with the order of protocols being randomized.

**Results**. In the regression analysis, there was a significant positive association between lactate delta changes immediately post-squat at 50% of maximum strength at session 2 with the variable "sex: women" ($\beta$: 3.02, 95% CI [−0.18–0.30], $p = 0.047$) and BMI ($kg/m^2$). Age exhibited a positive association ($\beta$: 0.19, 95% CI [0.02–0.36], $p = 0.032$) with lactate delta changes immediately post-squat at 75% of maximum strength at session 2. There was also a significant inverse association between lactate delta changes at 10 min post-squat test exercise at 75% of maximum strength at session 1 and 2, and vigorous physical activity (−0.01, 95% CI [−0.02–0.00], $p = 0.046$).

**Conclusion**. In summary, this study provides valuable insights into the association between lactate production and physical activity levels in young, healthy adults undergoing different squat training protocols. These findings suggest that intense physical activity may be associated with lower lactate production, indicating greater metabolic efficiency. In addition, sex differences in metabolic responses were observed, emphasizing the importance of personalized approaches in program design.

## INTRODUCTION

In recent decades, strength training has garnered increasing interest across various domains (*Brown, 2007*; *Hedrick, 1993*; *Zatsiorsky, Kraemer & Fry, 2020*). This extends beyond enhancing physical capacity and athletic performance, encompassing improvements in body composition, overall health, as well as stress and anxiety reduction (*Westcott, 2012*). Focusing on a foundational element of strength training, the squat assumes particular significance. It finds application not only in fitness conditioning programs but also in rehabilitation and recovery regimens for diverse injuries (*Comfort & Kasim, 2007*). Moreover, the squat is not confined to gymnasiums and training facilities (*Schlegel & Fialová, 2021*); it is a movement pattern replicated in everyday activities such as rising from and sitting in a chair, ascending, and descending stairs, picking objects from the ground, and more (*Lo, Kahya & Manor, 2022*). Therefore, understanding the physiological aspects of strength training, specifically pertaining to the squat exercise, could hold pivotal importance in optimizing strength training (*Hettinger, 2017*).

Given the importance of accurate and reliable measurement in strength training studies, the use of motorized resistance devices like the functional electromechanical dynamometer (FEMD) is crucial. The reliability of the FEMD has been previously confirmed, (*Baena-Raya et al., 2021*; *del-Cuerpo et al., 2023b*; *Reyes-Ferrada et al., 2021*; *Rodriguez-Perea et al., 2021*; *Sánchez-Sánchez et al., 2021*) demonstrating high test-retest reliability in different resistance exercises training protocols. Additionally, comparing the FEMD to other similar devices, such as the Sprint 1080, provides context for its selection and expected performance. The Sprint 1080 has been validated for measuring sprint performance with precision, highlighting its reliability in assessing sprint mechanics, acceleration, and deceleration (*Rakovic et al., 2022*). While the FEMD is optimal for controlled resistance training and rehabilitation, the Sprint 1080 excels in field settings for sprint performance assessments. The choice between these devices should be based on the specific requirements of the study, ensuring alignment with the research or training objectives.

Transitioning into the physiological realm, lactate is a metabolic marker intricately tied to physical exercise, which has been subject to meticulous scrutiny in the realm of endurance for many years (*Jones & Carter, 2000*). In fact, numerous articles assert that the exercise intensity corresponding to the increase in blood lactate above resting levels are potent predictors of performance in endurance training (*Beneke, Leithäuser & Ochentel, 2011*). On the other hand, the lactate threshold marks the point at which blood lactate concentrations begin to increase exponentially from resting values (*Davis, 1985*). As for the study of this variable during strength training, it has been examined to a lesser extent and not for as long, but there is indeed literature indicating its behavior (*Dominguez et al., 2018*; *Garnacho-Castaño, Dominguez & Maté-Muñoz, 2015*).

In connection with this, it becomes especially relevant to understand how lactate production varies at different intensities of strength training, with particular attention to the variation in prescription of repetitions and load percentage in the squat (*Garnacho-Castaño, Dominguez & Maté-Muñoz, 2015*). This study will generate new insights and shed light not only on lactate production following the completion of two protocols with

different squat training intensities but also on the relationship between lactate production and levels of physical activity in a group of young, healthy adults. This will provide a deeper understanding of the effects of different training protocols on the body's metabolic response.

Furthermore, assessing lactate levels during strength training and recovery is important because it provides valuable information about the metabolic response to exercise and the effectiveness of training sessions (*Garnacho-Castaño, Dominguez & Maté-Muñoz, 2015*). Lactate production is closely correlated with energy metabolism and can be used as a biomarker for fat oxidation in skeletal muscles (*Lee et al., 2023*). Monitoring lactate levels during training sessions can help coaches and athletes understand the level of adenosine triphosphate breakdown and the energy system contribution involved in muscle energy coverage during intense sprints (*Kantanista et al., 2021*). Additionally, lactate measurements can be used to estimate the maximal lactate steady state intensity, which represents the balance between lactate appearance and clearance in the bloodstream (*Messias et al., 2017*). Furthermore, lactate clearance during recovery periods can be indicative of the body's ability to recover and adapt to the training stimulus (*Lopes et al., 2014*). Overall, assessing lactate levels during strength training and recovery provides insights into energy metabolism, training effectiveness, and recovery capacity.

The uniqueness and relevance of this study lies in its ability to provide specific and applicable guidelines, supported by rigorous data and comprehensive analyses regarding the metabolic response to muscular strength training. Thus, the aim of this study was to determine the association between changes in lactate production and levels of physical activity in response to two squat training protocols (30 repetitions at 50% one repetition maximum (RM) and 12 repetitions at 75% 1RM) in a group of healthy young adults.

## MATERIALS & METHODS

### Subjects

Twenty-nine healthy subjects (age: 24.9 ± 4.6 years; height: 1.70 ± 0.1 m; body mass: 68.1 ± 12.9 kg; BMI: 23.5 ± 3.0 kg/m$^2$), including thirteen males and sixteen females, willingly participated in this study (Table 1). All participants met the eligibility criteria, which entailed (a) having no medical conditions and (b) having at least one year of experience in muscle strength training.

In the initial assessment to confirm eligibility, female participants were inquired about their menstrual cycle. This involved details like the start and end dates of their most recent menstruation, length of their menstrual cycle, any instances of intense discomfort or excessive bleeding, and use of hormonal contraceptives. We specifically assessed them during the luteal phase (*Bisdee, James & Shaw, 1989*). Additionally, none of them relied on hormonal contraceptives, and only two reported experiencing severe pain and heavy bleeding. All participants were informed of the nature, objectives, and risks associated with the experimental procedure. Subsequently, all of them provided their written consent and agreed to participate in the study. The study protocol was approved by the Committee on Human Research of the University of Granada (no. 2182/CEIH/2021) and was conducted in accordance with the Declaration of Helsinki.

**Table 1 Descriptive characteristics of the sample based on PAL.**

| 50th percentile | Total Mean (SD) | Low PAL (*n* = 15) Mean (SD) | High PAL (*n* = 14) Mean (SD) | Sig. | F value |
|---|---|---|---|---|---|
| Age | 24.93 (4.56) | 25.13 (4.10) | 24.71 (5.15) | 0.810 | 0.06 |
| Size | 1.70 (0.10) | 1.69 (0.10) | 1.71 (0.09) | 0.645 | 0.22 |
| Weight | 68.05 (12.92) | 69.02 (14.71) | 67.02 (11.14) | 0.685 | 0.17 |
| BMI | 23.48 (2.97) | 23.99 (3.30) | 22.93 (2.58) | 0.343 | 0.93 |
| Low-PAL (days/week) | 4.72 (2.85) | 4.20 (3.19) | 5.29 (2.43) | 0.314 | 1.05 |
| Low-PAL (minutes/week) | 37.93 (36.49) | 38.67 (41.21) | 37.14 (32.21) | 0.913 | 0.01 |
| Moderate-PAL (days/week) | 2.52 (2.32) | 2.13 (2.36) | 2.93 (2.30) | 0.367 | 0.84 |
| Moderate-PAL (minutes/week) | 66.03 (78.52) | 48.67 (64.43) | 84.64 (89.92) | 0.224 | 1.55 |
| Vigorous-PAL (days/week) | 3.69 (1.56) | 2.47 (1.13) | 5.00 (0.55) | 0.000 | 57.74 |
| Vigorous-PAL (minutes/week) | 90.52 (37.59) | 84.00 (45.44) | 97.50 (26.80) | 0.343 | 0.93 |

**Notes.**

PAL, physical activity level; SD, standard deviation.

## Intervention

A repeated measures design was employed to assess variations in energy expenditure (EE) during two different acute squat exercise protocols without arm swing using FEMD. Following one session of familiarization and 1RM determination, participants visited the lab four times within a two-week period, ensuring at least 48 h between visits. In each session, they completed three sets of 12 repetitions at 75% 1RM and three sets of 30 repetitions at 50% 1RM, with the order of protocols being randomized.

## Data collection procedures

During the study, participants underwent a total of five sessions, including one familiarization session and four experimental sessions. They were instructed to ensure a minimum of 8 h of sleep and to avoid smoking, engaging in vigorous exercise within 12 h before testing, and eating at least one hour prior to the session. To ensure uniform nutritional conditions among participants and minimize external influences on the results, all participants' diets were adjusted a week before and throughout the study. This involved the exclusion of potential influencing factors like caffeine and supplements. Participants arrived at the laboratory at the same time each day, within a one-hour window (*e.g.*, between 8:00 and 9:00 AM), and were exposed to similar environmental conditions, with a temperature of approximately 22 °C and a humidity level of 60%. Regular checks were performed to confirm that these conditions were consistent during each testing session

Participants followed the researcher's instructions upon arrival. As conducted in *del-Cuerpo et al., (2023b)* and *Del-Cuerpo et al. (2023a)*, they wore a gas analyzer mask and sat in a relaxed position for five minutes for initial gas analysis. At this time, capillary blood samples were collected from a finger and then analyzed. They then put on a vest equipped with a carabiner connected to an FEMD cable. After a 5-minute warm-up on a cycle ergometer, they did repetitions at 10% of their 1RM. Following a 5-minute rest, they performed three sets of 12 repetitions at 75% 1RM or 30 repetitions at 50% 1RM. Post-exercise gas analysis was conducted while seated for ten minutes. The exercise order

**Figure 1** **Protocol measurement of the squat exercise.** Image source credits: Drop of blood, from flaticon, https://www.flaticon.es/icono-gratis/silueta-de-gota-de-agua_34042; Warm-up on a cycle ergometer, from flaticon, https://www.flaticon.es/icono-gratis/bicicleta-estatica_538963?term=bicicleta+estatica&page=1&position=6&origin=search&related_id=538963. Squat exercise using DEMF: own elaboration. The figure was created by a professional photographer who took a picture of a person dressed in a special, completely black suit, squatting on a functional electromechanical dynamometer against a completely white background

was randomized using a computerized random number generator to ensure unbiased assignment. Each participant received a unique sequence of exercises, with the sequence generated prior to the study and kept confidential to avoid any influence on the testing procedures. This approach ensured effective and consistent randomization of the exercise order across all participants. Additionally, 5-minute breaks were implemented between sets. EE was determined indirectly using a metabolic cart that analyzed respiratory gases. For this purpose, the FitMate™ metabolic system (Cosmed, Rome, Italy) was employed, a reliable and validated analyzer designed to measure oxygen consumption and EE by analyzing breath-by-breath ventilation and expired oxygen and carbon dioxide levels (*Chmielewska, Kujawa & Regulska-Ilow, 2023*). Capillary blood samples were collected and analyzed immediately after the session and again 10 min post-exercise. Prior studies have confirmed the test-retest reliability of the FEMD for squat training (*del-Cuerpo et al., 2023b*). The study protocol is outlined in Fig. 1.

During their initial lab visit, participants engaged in a 60-minute session to familiarize themselves with the FEMD and establish their 1RM. This session consisted of (a) a general warm-up comprising 2 sets of 10 squat repetitions, starting with a load of 10 kg and increasing by 2 kg per repetition, with a 40-second rest between sets, and (b) a direct estimation of the participants' squat 1RM. The protocol used to determine the participants' 1RM, as described by *del-Cuerpo et al., (2023b)*, *Del-Cuerpo et al. (2023a)* began with a load of 100% of body weight for males and 80% of body weight for females, with increments of 4 kg (up to a maximum of 10 repetitions). Different outcomes followed: (1) If the participant was able to perform more than one repetition, they continued until failure. After this, there was a 5-minute rest, and the initial load was set at the maximum load that had been overcome, with increments of 1 kg until the resistance became insurmountable (up to a maximum of 5 repetitions). The last repetition performed was considered the 1RM of the participant. (2) If the participant could not perform any repetitions, there was a 2-minute rest, after which the initial load was set at 90% of body weight for males and 70% of body weight for females, with increments of 1 kg until the resistance became insurmountable (up to a maximum of 5 repetitions). The last repetition performed was considered the

1RM of the participant. (3) If the participant could only perform one repetition, there was a 5-minute rest, after which the initial load was the same as that started with, with increments of 1 kg until the resistance became insurmountable (up to a maximum of 5 repetitions). The last repetition performed was considered the 1RM of the participant. (4) If the participant's weight exceeded 120 kg, the total number of repetitions they could perform was observed, and the 1RM was estimated using a formula (*Vaquero-Cristóbal et al., 2020*).

## Measurements

A qualified Human Nutrition and Dietetics graduate tailored a weekly diet to each participant's energy requirements during this period. These requirements were determined through anthropometric measurements taken a week before the study and in subsequent weeks. These measurements encompassed weight (measured with a professional TANITA SC-240-MA scale with a biological suite), height (measured with a portable Seca 213 Stadiometer), and skinfold measurements for various areas, along with arm and mid-thigh circumferences, all taken by an ISAK level 1 anthropometrist using respective tools. Basal EE was computed using the Harris-Benedict formula (*Chmielewska, Kujawa & Regulska-Ilow, 2023*), total EE was determined using the appropriate activity factor, and body fat percentage was estimated using the Foulkner formula (*Jones et al., 2020*).

The half squat exercise without arm swing was conducted using the FEMD (Dynasystem, Model Research, Granada, Spain), a validated isokinetic multi-joint device capable of assessing various strength parameters such as speed, power, work, and impulse in one instrument (*del-Cuerpo et al., 2023b*; *Rodriguez-Perea et al., 2021*). Each participant's physical activity level (PAL) was evaluated using the International Physical Activity Questionnaire (IPAQ) short form, a reliable tool for assessing physical activity in adults aged 18 to 69 years (*Roman-Viñas et al., 2010*).

In order to measure lactate production, blood samples were collected from the index finger of each participant using sterile single-use safety lancets (Accu-Check Safe-T-Pro Plus, Indianapolis, IN, USA). Before collection, the site was cleaned with a sterile alcohol swab in a circular motion and allowed to air dry completely. The site was then lanced with a sterile single-use lancet. Test strips inserted into a reliable and valid device: Lactate Pro 2 (Arkray, Kyota, Japan) (*Crotty et al., 2021*) were then sequentially placed against the surface of this blood sample until beeps were audible, signifying the start of analysis. The same operator collected and analyzed all blood samples throughout the study.

For measuring EE during both protocols, the FitMate™ metabolic system (Cosmed, Rome, Italy) was employed. This is a dependable and validated compact metabolic analyzer designed to measure oxygen consumption and EE during both rest and exercise, analyzing breath-by-breath ventilation as well as expired oxygen and carbon dioxide levels (*Chmielewska, Kujawa & Regulska-Ilow, 2023*). It ensures accurate measurements through several advanced features, including a turbine flowmeter for measuring breath-by-breath ventilation and a galvanic fuel cell oxygen sensor for analyzing the fraction of oxygen in expired gases. Additionally, the FitMate™ incorporates a patent-pending innovative sampling technology that preserves the performance of a traditional metabolic cart with a

standard mixing chamber. The system continuously monitors environmental conditions such as humidity, temperature, and barometric pressure, which are utilized in internal calculations. Oxygen uptake is calculated using standard metabolic formulas, while EE is determined with a fixed respiratory quotient (RQ) of 0.85 (*Nieman et al., 2006*).

The device self-calibrates before testing each subject and does not require a warm-up period. During the study, the mask was worn by the participant for ten minutes after completing the exercise. In case of poor fit, the device provided a warning on the screen. The system also includes a sample line connected to a 3-way valve in the Douglas bag system, allowing simultaneous sampling of expired air from the subjects. These features ensured that the utilization of this device did not interfere with the execution of the squat protocol and contributed to its reliability in measuring EE in our study (*Nieman et al., 2006*).

### Statistical analyses

Statistical analysis was performed using SPSS$^{®}$ v23.0 software (IBM Corp., Armonk, NY, USA). Data are presented as the mean and standard deviation (SD). Normality and homoscedasticity assumptions for all data were checked using the Shapiro–Wilk and Levene tests. Differences between groups according to PAL were determined using the Student *t*-test. Delta changes were determined by comparing Pre *vs* ImmPost test and after 10 min pos*t*-test. To investigate the association of lactate delta changes with sex, age, BMI, and PAL, a multivariable regression was conducted with results reported as beta coefficient (b) and their 95% CI. The sample size for this experimental study was determined using statistical software (G*Power version 3.0.10), based on a test power of 90% and a statistical significance level of 5%, with an effect size of 0.8.

## RESULTS

Descriptive characteristic of the subjects participating in this study depending on their PAL is shown in Table 1.

In the comparison based on PAL (Table 2), the High-PAL reported lower delta change in Pre *vs* ImmPost test at 75% 1RM session 1 (S1) (Low-PAL 5.66 ± 2.86 *vs.* High-PAL 3.69 ± 2.31, $P = 0.050$) and delta $Post_{10min}$ at 50% 1RM S1 (Low-PAL 6.52 ± 3.28 *vs.* 4.11 ± 2.26, $P = 0.030$).

In the regression analysis, there was a significant positive association between lactate delta changes immediately post-squat at 50% 1RM session 2 (S2) with the variable "sex (biological attributes that are associated with physical and physiological features): women" ($β$: 3.02, 95% CI [−0.18–0.30], $p = 0.047$) and BMI ($kg/m^2$). Age exhibited a positive association ($β$: 0.19, 95% CI [0.02–0.36], $p = 0.032$) with lactate delta changes immediately post-squat at 75% 1RM S2 (Table 3).

In the regression model, there was a significantly inverse association between lactate delta changes 10 min post-squat test exercise at 75% 1RM S1 and S2 and vigorous physical activity (−0.01, 95% CI [−0.02–0.00], $p = 0.046$) (Table 4).

**Table 2  Delta change immediately and 10 minutes post-squat exercise.**

| 50th percentile | Low-PAL Mean (SD) | High-PAL Mean (SD) | Sig. | F value |
|---|---|---|---|---|
| Delta ImmPost 50% 1RM S1 | 8.73 (0.73) | 6.86 (2.70) | 0.097 | 2.95 |
| Delta ImmPost 50% 1RM S2 | 8.85 (3.12) | 6.89 (2.36) | 0.068 | 3.62 |
| Delta ImmPost 75% 1RM S1 | 5.66 (2.86) | 3.69 (2.31) | 0.050[*] | 4.15 |
| Delta ImmPost 75% 1RM S2 | 4.16 (2.26) | 3.94 (1.61) | 0.770 | 0.09 |
| Delta Post$_{10min}$ 50% 1RM S1 | 6.52 (3.28) | 4.11 (2.26) | 0.030[*] | 5.25 |
| Delta Post$_{10min}$ 50% 1RM S2 | 6.27 (6.27) | 4.39 (2.44) | 0.059 | 3.88 |
| Delta Post$_{10min}$ 75% 1RM S1 | 2.40 (2.40) | 1.53 (1.05) | 0.107 | 2.79 |
| Delta Post$_{10min}$ 75% 1RM S2 | 2.23 (2.23) | 1.64 (1.05) | 0.201 | 1.72 |

Notes.
[*]$p \leq 0.05$.
ImmPost: immediately post; 1RM: one repetition maximum; S1: session 1; S2: session 2.

**Table 3  Association of lactate delta changes immediately post-squat test with PAL.**

| | 50% 1RM S1 | | | 50% 1RM S2 | | |
|---|---|---|---|---|---|---|
| | β(95% CI) | Beta (SE) | P-value | β(95% CI) | Beta (SE) | P-value |
| Age (years) | 0.08 (−0.19; 0.34) | 0.11 (0.13) | 0.559 | 0.06 (−0.18; 0.30) | 0.09 (0.12) | 0.617 |
| Sex (men ref) | −0.79 (−4.09; 2.51) | −0.13 (1.60) | 0.627 | 3.02 (0.05; 5.99) | 0.53 (1.44) | 0.047[*] |
| BMI (kg/m$^2$ ) | 0.30 (−0.50; 1.10) | 0.29 (0.39) | 0.447 | 0.97 (0.25; 1.69) | 0.99 (0.35) | 0.011[*] |
| Body fat (kg) | 0.07 (−0.36; 0.50) | 0.12 (0.21) | 0.750 | −0.38 (−0.77; 0.01) | −0.69 (0.19) | 0.055 |
| Vigorous PAL (MET) | 0.00 (0.00; 0.00) | −0.21 (0.00) | 0.282 | 0.00 (0.00; 0.00) | −0.22 (0.00) | 0.223 |
| | 75% 1RM S1 | | | 75% 1RM S2 | | |
| | β(95% CI) | Beta (SE) | P-value | β(95% CI) | Beta (SE) | P-value |
| Age (years) | −0.21 (−0.46; 0.05) | −0.34 (0.12) | 0.111 | 0.19 (0.02; 0.36) | 0.44 (0.08) | 0.032[*] |
| Sex (men ref) | −2.41 (−5.60; 0.78) | −0.44 (1.54) | 0.132 | 0.28 (−1.83; 2.39) | 0.07 (1.02) | 0.785 |
| BMI (kg/m$^2$ ) | −0.23 (−1.01; 0.54) | −0.25 (0.37) | 0.545 | 0.34 (−0.17; 0.85) | 0.52 (0.25) | 0.186 |
| Body fat (kg) | 0.21 (−0.20; 0.63) | 0.41 (0.20) | 0.302 | −0.13 (−0.41; 0.14) | −0.36 (0.13) | 0.328 |
| Vigorous PAL (MET) | 0.00 (0.00; 0.00) | −0.17 (0.00) | 0.401 | 0.00 (0.00; 0.00) | −0.13 (0.00) | 0.481 |

Notes.
[*]$p \leq 0.05$.
95% CI: 95% confidence interval; SE: standard error; PAL: physical activity level.

## DISCUSSION

The main purpose of this study was to determine the association between changes in lactate production and levels of physical activity in response to two squat training protocols (30 repetitions at 50% 1RM and 12 repetitions at 75% 1RM) in a group of healthy young adults. The obtained results indicate that (1) subjects with higher PAL report lower lactate production during squat exercises at 50% and 75% 1RM, (2) women exhibit an association with higher lactate production than men in squat exercises, and (3) higher age is correlated with greater lactate production.

Before delving into the obtained results, conducting an in-depth literature review, we observed that most articles discussing blood lactate concentration or lactate clearance and its behavior in trained or sedentary subjects date from the 1970s to around 2010. There are

**Table 4  Association of lactate delta changes 10 min post-squat test with PAL.**

| | 50% 1RM S1 | | | 50% 1RM S2 | | |
|---|---|---|---|---|---|---|
| | β(95%CI) | Beta (SE) | P-value | β(95%CI) | Beta (SE) | P-value |
| Age (years) | 0.18 (−0.08; 0.44) | 0.27 (0.13) | 0.170 | 0.05 (−0.19; 0.28) | 0.08 (0.11) | 0.692 |
| Sex (men ref) | −0.1 (−3.35; 3.12) | −0.02 (1.57) | 0.942 | 1.63 (−1.31; 4.57) | 0.31 (1.42) | 0.262 |
| BMI (kg/m$^2$ ) | 0.35 (−0.44; 1.13) | 0.34 (0.38) | 0.372 | 0.47 (−0.25; 1.18) | 0.52 (0.35) | 0.189 |
| Body fat (kg) | −0.03 (−0.45; 0.40) | −0.05 (0.20) | 0.895 | −0.19 (−0.57; 0.20) | −0.37 (0.19) | 0.321 |
| Vigorous PAL (MET) | 0.00 (0.00; 0.00) | −0.32 (0.00) | 0.096[*] | −0.01 (−0.02; 0.00) | −0.39 (0.00) | 0.046[*] |
| | 75% 1RM S1 | | | 75% 1RM S2 | | |
| | β(95%CI) | Beta (SE) | P-value | β(95%CI) | Beta (SE) | P-value |
| Age (years) | 0.01 (−0.14; 0.15) | 0.02 (0.06) | 0.913 | 0.07 (−0.03; 0.17) | 0.25 (0.05) | 0.176 |
| Sex (men ref) | 0.55 (−1.23; 2.33) | 0.19 (0.86) | 0.532 | 0.14 (−1.09; 1.36) | 0.06 (0.59) | 0.821 |
| BMI (kg/m$^2$ ) | 0.25 (−0.18; 0.69) | 0.52 (0.21) | 0.236 | 0.27 (−0.03; 0.57) | 0.65 (0.14) | 0.075[*] |
| Body fat (kg) | −0.10 (−0.33; 0.13) | −0.36 (0.11) | 0.393 | −0.06 (−0.06; 0.22) | −0.27 (0.08) | 0.424 |
| Vigorous PAL (MET) | 0.00 (0.00; 0.00) | −0.08 (0.00) | 0.704 | 0.00 (0.00; 0.00) | −0.13 (0.00) | 0.131 |

**Notes.**
[*]$p \leq 0.05$.
95% CI: 95% confidence interval; SE: standard error; PAL: physical activity level.

few articles investigating this aspect today. Furthermore, while lactate production during training has been more extensively investigated in endurance training to the best of our knowledge, there are also some studies, albeit to a lesser extent, that examine it during strength training. It would be interesting to have more current articles that evaluate this variable in subjects with different PAL. However, we believe that this is precisely what gives uniqueness to our research.

As a primary finding of this study, we observed a strong association between self-reported PAL and lactate production during squat exercise. Our results align with those published by *Hu et al. (2009)*, who sought to assess the effects of strength training on work capacity and parasympathetic heart rate modulation during exercise in physically inactive men. For this purpose, they compared a sedentary control group with an experimental group that underwent 10 weeks of progressive strength training. A decrease in blood lactate levels at submaximal intensities was observed compared to the control group. This is due to adaptations involving greater lactate removal and oxidation in active muscles (*Messonnier et al., 2013*). It has been shown that training alters the activity of lactate dehydrogenase by shifting its distribution toward a higher proportion of its H-LDH isoenzyme (*Messonnier et al., 2001*), which is more favorable for lactate oxidation to pyruvate than the M-LDH isoenzyme. Training has also been shown to increase muscle oxidative capacity and lactate oxidation through enhancements in mitochondrial mass and the expression of mitochondrial constituent proteins (*Hood et al., 2011*; *Pesta et al., 2011*).

Regarding to sex association to lactate production, despite observing that in three out of four sessions there is no association between sex and lactate production, men still exhibit higher lactate levels. Conversely, we do find a significant and positive association between lactate production and female in the second session at 50% of 1RM, but this is not the case in the rest of the sessions. The evidence found demonstrates that men

have higher lactate production. Specifically, *Mochizuki et al. (2022)* reported that blood lactate concentration immediately after exercise was significantly higher in men than in women, although these results were not significant. On the other hand, *Szivak et al. (2013)* indicated that blood lactate concentration was significantly higher in men than in women after strength training. This might be attributed to a constraining phosphorylase function in women, possibly stemming from either a reduced maximal velocity or an elevated Michaelis constant for the enzyme, or diminished levels of alternative activators like cAMP (*Esbjornsson-Liljedahl et al., 1999*). These results contrast with the significant association observed in the second session at 50% of 1RM obtained in our article but align with the results obtained in the other three sessions where there is no association between sex and lactate production.

Regarding the relationship between age and lactate production, research has shown that lactate production tends to be higher in older individuals. Our findings are supported by those published by *Jansson et al. (1994)*, who stated many years ago that lactate levels in obese men were significantly higher than those in lean males. This is because white adipocytes produce large amounts of lactate (*Petersen et al., 2017*). Additionally, in a more recent study, *Okano et al. (2022)* indicated that for every incremental increase in body fat ratio, a 0.157 mmol/L increase in blood lactate level can be expected ($p = 0.001$), further reinforcing the positive relationship between BMI and lactate production.

Finally, older adults tend to exhibit higher lactate production during and after exercise compared to their younger counterparts. To the best of our knowledge, there are no papers comparing lactate production between young and older adults during and after resistance training. However, despite this, there are a few articles that corroborate our findings. Firstly, *Seals et al. (1984)* found that endurance training resulted in adaptations in the blood lactate response to submaximal exercise in older individuals. *Masuda et al. (2009)* suggested that aging causes metabolic changes in skeletal muscle that can reduce lactate accumulation during exercise and increase fatigue resistance. *Tzankoff & Norris (1979)* observed an age-related decrease in the ability to diffuse lactate from muscle and distribute it into its space, potentially affecting endurance and work capacity. Finally, *Hermansen (1971)* highlighted lactate production as an indicator of anaerobic processes during exercise. Collectively, these papers suggest that age-related physiological changes in muscle metabolism contribute to higher lactate production in older adults during and after exercise.

Despite the significant findings of this study, it is important to acknowledge some limitations that may influence the interpretation of the results. Firstly, the sample consisted exclusively of young, healthy adults, limiting the generalizability of the findings to populations with different demographic characteristics. Additionally, the study specifically focused on two squat training protocols, which may not fully reflect the complexity of metabolic responses in other exercises or training modalities.

The results of this study have practical implications for strength training prescription in different populations. Firstly, they suggest that individuals with higher levels of physical activity may experience an improvement in lactate metabolism efficiency, which can have a positive impact on performance and recovery. This supports the importance of including

strength training in physical activity programs to promote beneficial metabolic adaptations. Additionally, the identification of sex disparities in lactate production highlights the need to consider differentiated training strategies for men and women.

The findings from this study have significant clinical implications, especially for populations undergoing rehabilitation or dealing with metabolic disorders. Lower lactate production in individuals with higher physical activity levels suggests that structured strength training programs could be beneficial for improving metabolic and cardiovascular health, facilitating the rehabilitation of musculoskeletal injuries, and enhancing metabolic efficiency in older adults. Additionally, the observed sex differences in lactate production highlight the need for specific and personalized clinical interventions, potentially leading to more effective and tailored rehabilitation and training programs for both men and women.

## CONCLUSIONS

In summary, this study provides valuable insight into the association between lactate production and PAL in young, healthy adults undergoing different squat training protocols. The findings suggest that intense physical activity may be related to lower lactate production, indicating greater metabolic efficiency. Additionally, sex differences in metabolic response were observed, emphasizing the importance of personalized approaches in program design. These results contribute to the growing body of knowledge in the field of strength training and provide useful information for optimizing physical performance in diverse populations. However, it is crucial to consider the study's limitations when interpreting and applying these findings. Future research could address these limitations and continue to expand our understanding of metabolic responses to strength training.

## ACKNOWLEDGEMENTS

This article will be part of Indya del-Cuerpo's doctoral thesis performed in the Biomedicine Doctorate Program of the University of Granada.

### Funding

The postdoctoral researcher Pedro Delgado-Floody has a contract through the programme "Recualificación del Profesorado Universitario. Modalidad María Zambrano", Universidad de Granada/Ministerio de Universidades y Fondos Next Generation de la Unión Europea and Daniel Jerez-Mayorga has a contract through the programme "Recualificación del Profesorado Universitario. Modalidad Margarita Salas", Universidad de Granada/Ministerio de Universidades y Fondos Next Generation de la Unión Europea. This work was supported by Spanish Ministry of Universities (FPU19/02030), and the High Council for Sports (CSD); Spanish Ministry of Culture and Sports (09/UPB/23), and the project DIE22-0007, Universidad de Granada. The funders had no role in study design, data collection and analysis, decision to publish, or preparation of the manuscript.

## Grant Disclosures

The following grant information was disclosed by the authors:

Recualificación del Profesorado Universitario. Modalidad María Zambrano.

Recualificación del Profesorado Universitario. Modalidad Margarita Salas.

Universidad de Granada/Ministerio de Universidades y Fondos Next Generation de la Unión Europea.

Spanish Ministry of Universities: FPU19/02030.

High Council for Sports (CSD).

Spanish Ministry of Culture and Sports: 09/UPB/23.

Consejería de Universidad, Investigación e Innovación and by ERDF Andalusia Program 2021-2027: A.SEJ.227.UGR23.

## Competing Interests

The authors declare there are no competing interests.

## Author Contributions

- Indya del-Cuerpo conceived and designed the experiments, performed the experiments, prepared figures and/or tables, authored or reviewed drafts of the article, and approved the final draft.
- Daniel Jerez-Mayorga performed the experiments, analyzed the data, prepared figures and/or tables, authored or reviewed drafts of the article, and approved the final draft.
- Luis Javier Chirosa-Ríos performed the experiments, authored or reviewed drafts of the article, and approved the final draft.
- Felipe Caamaño-Navarrete performed the experiments, authored or reviewed drafts of the article, and approved the final draft.
- Pedro Delgado-Floody conceived and designed the experiments, analyzed the data, authored or reviewed drafts of the article, and approved the final draft.

## Human Ethics

The following information was supplied relating to ethical approvals (i.e., approving body and any reference numbers):

The study protocol was approved by the Committee on HumanResearch of the University of Granada (no. 2182/CEIH/2021) and was conducted in accordance with the Declaration of Helsinki.

## Data Availability

The raw data are available in the Supplemental File.

## Supplemental Information

Supplemental information for this article can be found online at http://dx.doi.org/10.7717/peerj.18215#supplemental-information.

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
