# Peer review of "The influence of physical activity levels on lactate production during squat training using a functional electromechanical dynamometer"

_PeerJ, doi:10.7717/peerj.18215_

## Round 0.1 · original submission · Minor Revisions

Several reviewers have made reasonable suggestions. Please consider them and revise the manuscript accordingly.

Reviewer 1 ·

Basic reporting

In my opinion, the study topic is not quite new one. The study not well designed, and the subject's selection could not accept because of age variations and gender differences in mentioned ages (mainly for ages 10-11). In these ages, there is a significant difference between boys and girls. The Reeson for this is that the girls may get experience growth spot.
Using male and female words are not recommended for these ages; it should be noted that boys and girls words are quite suitable as well as athlete should not be used because the children should be in sport to fun and enjoy Not train like athletes.
Scientific background not quite strong and have huge misunderstanding. There seems to be many of sentences have been stated based on personal experience of idea NOT valuable citations or cited the old sources that these facts are modified nowadays.

Experimental design

As I mentioned above,
subjects,
research design,
study criteria,
grouping
have huge misunderstanding.

Validity of the findings

No comments

Additional comments

No comments

·

Basic reporting

I have gone through this paper and it is well organized, and followed the manuscript guidelines. The authors have really done well and I am confident with their work. The introduction section is well structured and shows the importance of the study.

Experimental design

The research questions are well defined, relevant & meaningful. It is evident that authors have done rigorous investigation and the method used is commendable. However, if I may suggest, it would be better if not best to specify under the method & intervention if the squats used are done with or without arm swing.

Validity of the findings

The outcomes of the study are consistent with the findings.

Additional comments

In my opinion, the presentation of thoughts in the paper is notable. It will make a contribution to the relevant field of research.

Reviewer 3 ·

Basic reporting

Dear authors,

Congratulations on your article. This study is very interesting; however, minor revisions can improve the content of your article. Below are some detailed suggestions for each section:

Introduction:
Please discuss the reliability of the motorized resistance device used in your study. It would be beneficial to compare this device with other similar devices, such as the Sprint 1080, in terms of reliability and related athletic uses. Providing a comparative analysis will help to contextualize the choice of equipment and its expected performance in your study.

Methods:

Please elaborate on the methodology used to calculate the one-repetition maximum (1RM). A detailed explanation will enhance the reader's understanding of the rigor and accuracy of your measurements.
Discuss in greater detail how you calculated the sample size. It would be more precise to identify the minimum sample size needed using data from previous studies to determine effect sizes in at least one outcome measure. Including this information will strengthen the validity of your study's design and findings.
Figure 1 is appropriate, but it would be very helpful for the reader to include real images from the measurement procedures. This visual aid will provide a clearer understanding of the experimental setup and enhance the reproducibility of your study.
Discussion:
The discussion section is organized appropriately. However, it would be useful to include the clinical implications of your findings. You have included practical implications related to strength training and fitness, but please discuss whether this study holds clinical implications for clinical populations as well. Highlighting potential clinical applications will broaden the impact of your research and underscore its relevance to a wider audience.

Thank you for considering these revisions. I believe they will enhance the clarity and impact of your article. Best of luck with your manuscript.

Experimental design

No comment

Validity of the findings

Bo comment

Additional comments

No comment

Reviewer 4 ·

Basic reporting

This manuscript is well written in English.

Experimental design

Most important information was well written in experimental design as well. However, there was some duplicated information in two or three paragraphs. If authors revised those paragraphs, it will be clear and professional to readers.
Line 137: EE just came out without expression of full name previously.
Line 180-184: New information just came out here. it could be added in Line 146-151.
Line 153-176: Some information is overlapped so re-paraphasing is required to be clear.

Validity of the findings

The results are reported well.
Line 286: I did not realized that research was interested in influence of sex. Why did you look at this?

In general, the discussion needs to be developed with more information using detail mechanisms.

Additional comments

The title includes motorized resistance device. I could not find any where that contains any information regarding motorized resistance device.
If you used the motorized resistance device, research might assume that there is some differences between traditional resistance device and motorized one.
The above comments have to be addressed.

Reviewer 5 ·

Basic reporting

The introduction provides a clear and comprehensive overview of the study's context, effectively outlining the importance of examining lactate production during squat training with a motorized resistance device. The experimental procedures are meticulously detailed and well-executed, ensuring methodological rigor and transparency in data collection. The statistical analysis is robust and appropriately applied, effectively addressing the research hypotheses and ensuring the reliability of the study's conclusions. The manuscript is written in clear, professional English, maintaining a high standard of readability and scientific communication. The literature references provide a thorough background and context for the study, demonstrating a comprehensive understanding of relevant research in the field. The article is well-structured with logically organized sections, enhancing readability and comprehension. The figures and tables are well-designed and effectively support the text, providing visual clarity and aiding in the interpretation of results. The results are directly aligned with the study's hypotheses, demonstrating a clear connection between the research questions posed and the findings obtained. Overall, the manuscript showcases a commendable adherence to scientific standards, with clear methodology, rigorous analysis, and a strong focus on relevance and clarity.

Experimental design

However, prior to publication, I suggest some alterations which I hope the authors will consider. By addressing these points, the methodology section will be clearer and more precise for readers.
Line 123-124 of your manuscript. How did you determine the sample size for your study? Was it based on a specific statistical power analysis, and if so, what were the parameters used (e.g., effect size, significance level, power)?
What rationale or criteria did you use to arrive at the number of participants (29 healthy subjects, including 13 males and 16 females)?
Line 130 of your manuscript. Why did you choose to specifically assess the female participants during the luteal phase of their menstrual cycle, and how do you anticipate this choice impacting your study’s results?
Line 138-143 of your manuscript. What was the rationale behind choosing the specific squat exercise protocols (3 sets of 12 repetitions at 75% 1RM and 3 sets of 30 repetitions at 50% 1RM) for assessing energy expenditure (EE)? Could you elaborate on the method used for randomizing the order of protocols, and how you ensured this randomization minimized potential biases in the results? How did you determine that the familiarization session was sufficient for participants to perform the exercises correctly and consistently during the actual testing sessions?
Line 151: Specify the "one-hour window" for arrival to ensure it is clear (e.g., between 8-9 AM).
Lines 151-152: Clarify if environmental conditions (temperature and humidity) were monitored and maintained throughout all sessions.
Lines 154-157: Ensure the description of the gas analysis procedure is succinct and clear. For example, "Participants wore a gas analyzer mask and sat for five minutes for initial gas analysis, during which capillary blood samples were collected."
Line 161: Specify how the exercise order was randomized to provide clarity on the randomization process.
Line 162: Briefly describe the metabolic cart used for indirect EE determination to provide context on the equipment's reliability.
Line 202: Could you clarify the procedures followed for cleaning the site prior to blood collection using the sterile lancets? Providing details on the cleaning method would ensure consistency and accuracy in sample collection.
Regarding lines 208-215: Could you clarify how the FitMateô metabolic system ensured accurate measurements of energy expenditure (EE), particularly regarding the handling of breath-by-breath ventilation and calibration procedures? Providing specifics on these aspects would enhance understanding of the device's reliability inyour study.

Validity of the findings

Overall, the manuscript showcases a commendable adherence to scientific standards, with clear stated.

Additional comments

However, prior to publication, I suggest some alterations which I hope the authors will consider. By addressing these points, the methodology section will be clearer and more precise for readers.
Line 123-124 of your manuscript. How did you determine the sample size for your study? Was it based on a specific statistical power analysis, and if so, what were the parameters used (e.g., effect size, significance level, power)?
What rationale or criteria did you use to arrive at the number of participants (29 healthy subjects, including 13 males and 16 females)?
Line 130 of your manuscript. Why did you choose to specifically assess the female participants during the luteal phase of their menstrual cycle, and how do you anticipate this choice impacting your study’s results?
Line 138-143 of your manuscript. What was the rationale behind choosing the specific squat exercise protocols (3 sets of 12 repetitions at 75% 1RM and 3 sets of 30 repetitions at 50% 1RM) for assessing energy expenditure (EE)? Could you elaborate on the method used for randomizing the order of protocols, and how you ensured this randomization minimized potential biases in the results? How did you determine that the familiarization session was sufficient for participants to perform the exercises correctly and consistently during the actual testing sessions?
Line 151: Specify the "one-hour window" for arrival to ensure it is clear (e.g., between 8-9 AM).
Lines 151-152: Clarify if environmental conditions (temperature and humidity) were monitored and maintained throughout all sessions.
Lines 154-157: Ensure the description of the gas analysis procedure is succinct and clear. For example, "Participants wore a gas analyzer mask and sat for five minutes for initial gas analysis, during which capillary blood samples were collected."
Line 161: Specify how the exercise order was randomized to provide clarity on the randomization process.
Line 162: Briefly describe the metabolic cart used for indirect EE determination to provide context on the equipment's reliability.
Line 202: Could you clarify the procedures followed for cleaning the site prior to blood collection using the sterile lancets? Providing details on the cleaning method would ensure consistency and accuracy in sample collection.
Regarding lines 208-215: Could you clarify how the FitMateô metabolic system ensured accurate measurements of energy expenditure (EE), particularly regarding the handling of breath-by-breath ventilation and calibration procedures? Providing specifics on these aspects would enhance understanding of the device's reliability inyour study.

---

## Round 0.2 · accepted · Accept

Ready for publication. no comments remaining.

Reviewer 3 ·

Basic reporting

Accept

Experimental design

Accept

Validity of the findings

Accept

Reviewer 5 ·

Basic reporting

The revised manuscript is written in clear, professional English, maintaining a high standard of readability and scientific communication. It can be consider for publication.

Experimental design

Written good

Validity of the findings

Written good

Additional comments

The manuscript presents a well-structured and thoroughly researched study that effectively addresses an important aspect of lactate production during squat training. The authors have demonstrated a strong understanding of the subject matter, supported by robust methodology and rigorous statistical analysis. The clear and concise writing, along with well-organized sections, enhances the manuscript's readability and scientific merit. Overall, this study is a valuable contribution to the field and is recommended for publication.